# Relationship between dietary factors and the number of altered metabolic syndrome components in Chinese adults: a cross-sectional study using data from the China Health and Nutrition Survey

Maowei Cheng,[1,2] Huijun Wang,[1] Zhihong Wang,[1] Wenwen Du,[1] Yifei Ouyang,[1] Bing Zhang[1]

► Prepublication history and additional material are available. To view these files please visit the journal online (http://dx.doi.org/10.1136/bmjopen-2016-014911).

[1]National Institute for Nutrition and Health, Chinese Center for Disease Control and Prevention, Beijing, China
[2]Center for Disease Control and Prevention of Hubei Province, Wuhan, China

**Correspondence to**
Dr Bing Zhang;
zzhangb327@aliyun.com

## ABSTRACT

**Objectives** To study the correlation between dietary factors and the number of altered metabolic syndrome components (MetS) in Chinese adults systematically.

**Setting** A cross-sectional study using demographic and dietary data of adults aged 18–75 years from the China Health and Nutrition Survey (2009) was conducted in nine provinces in China.

**Participants** There were 6034 eligible subjects (2800 men and 3234 women) in this study.

**Outcomes** The primary outcome of this study were diet assessments and the number of altered MetS components. Dietary intake was measured using a combination of a 3-day period with 24-hour and household food inventory; average daily intakes of nutrients were estimated according to the Chinese Food Composition Table. Blood samples were analysed in a national central laboratory and the number of clustering MetS components was calculated by adding the presence of each MetS component.

**Results** After adjusting for covariates, and taking zero MetS as comparison, the high risk factors correlating with increased numbers of altered MetS components in men were higher intake of protein (70.4–73.4 g; $P$trend=0.0004), cholesterol (238.7–266.6 mg; $P$trend=0.004), meat (90.6–105.7 g; $P$trend=0.016), fish/seafood (30.4–42.3 g; $P$trend=0.001), and lower intake of coarse cereals (16.5–12.7 g; $P$trend=0.051), tubers (37.3–32.7 g; $P$trend=0.030), and dietary fibre (11.7–11.5 g; ANCOVA p=0.058). Meanwhile, the high risk factors correlating with the increased number of altered MetS components in women were higher intake of wheat (101.9–112.6 g; $P$trend=0.066) and sodium (3862.3–4005.7 mg, $P$trend=0.032), and lower intake of β-carotene (1578.6–1382.7 µg; $P$trend=0.007), milk, and dairy products (17.8–11.5 g; $P$trend=0.002).

**Conclusions** Some foods and nutritional factors correlate with an increased number of altered MetS components in Chinese adults. More prospective, multicentre and clinical research work to further examine these associations is underway.

### Strengths and limitations of this study

► This is the first observational study to examine the relationship between dietary factors and the number of altered MetS components among Chinese adults in a systematic way.

► This study includes a large sample size that is fairly representative of the Chinese population. Furthermore, only healthy adults that were not using chronic medication were included.

► In this study, we measured diet using a combination of 3 consecutive days by 24-hour dietary recall and a household inventory. One of the prominent advantages of this method is that it combines 24-hour individual recall with edible oil and other condiment consumption measured by household inventory, which greatly improves the quality of individual dietary data.

► The major limitation of this study is our inability to come to any causal conclusion due to the cross-sectional design.

## INTRODUCTION

Metabolic syndrome (MetS) is a cluster of risk factors (dysglycaemia, elevated blood pressure, dyslipidaemia, and central adiposity) that has been shown to be predictive of cardiovascular disease and diabetes.[1]

The overall age-standardised estimate of MetS prevalence in Chinese adults is 18.2%, according to the International Diabetes Federation, and 21.3%, according to the modified Adult Treatment Panel III criteria of the National Cholesterol Education Program (NCEP-ATP III). These estimates of MetS prevalence increase gradually with age.[2] Therefore, MetS is considered to be a major challenge that affects the quality of life of

millions of people, and there is an urgent need to identify effective strategies to better control MetS.

MetS requires three of five components: waist circumference (WC), fasting glucose, high density lipoprotein cholesterol (HDL-C), triglycerides, and hypertension.[1] MetS in individual patients may involve the alteration of various combinations of these components. Accumulating evidence has suggested that an increased number of altered MetS components is strongly associated with a higher risk of developing diseases.[3–6] Serum magnesium level decreased as the number of altered MetS components increased in elderly Greek patients.[5] The number of altered MetS components in middle-aged and elderly Japanese patients was related to lower intake of vitamin B6[6] and lower dietary fibre in men, and to lower intake of calcium, milk and dairy products, and higher cereal intake in women, after adjusting for age, energy intake, alcohol intake, smoking status, and physical activity.

An increasing number of studies in China have focused on the influence of dietary patterns on the prevalence of MetS and its altered components. Using data from the 2002 China National Nutrition and Health Survey (CNNHS), three types of dietary patterns have been identified in the Chinese population. One dietary pattern is called 'Yellow Earth' or 'traditional northern' pattern; it is characterised by consumption of refined cereal products, salted vegetables and tubers. Another dietary pattern is called 'Western/new affluence'; it is characterised by consumption of beef, lamb, milk, cheese, and yoghourt. The third dietary pattern is called 'Green Water' or 'traditional southern' pattern; it is characterised by consumption of rice, vegetables, and moderate amounts of animal foods.[7–9] The 'Yellow Earth' pattern and the 'Western/new affluence' pattern were associated with an increased likelihood of MetS, or with increased alterations of its components,[8 9] in comparison to the 'Green Water' dietary pattern. Our previous studies have identified a modern high-wheat pattern that is characterised by consumption of wheat products, nuts, fruits, eggs, milk and instant noodles or frozen dumplings. This modern pattern could be considered as a combination of the 'Yellow Earth' and 'Western/new affluence' patterns, and has been positively associated with diabetes.[10 11] Additionally, in our previous studies an 'alcohol' dietary pattern was identified among men. This dietary pattern is characterised by consumption of meat, alcohol and nuts, and has been positively associated with overweight/obesity and central obesity.[12]

However, it would be difficult to obtain an identical relationship between dietary patterns (or dietary index) and the relevant disease outcomes in different studies.[13] More importantly, because diet is a complex variable, it is necessary to adopt multiple approaches to examine the relationship between diet and the risk of diseases. If there were any effects caused by a particular nutrient, the dietary pattern approach would not be optimal to examine the relationship between diet and risk of diseases because the effect of that nutrient would be diluted by other factors.[14]

There are significant differences in the dietary characteristics of men and women.[15 16] To our knowledge, the nutrients that could have a positive effect on MetS or on its components have not been determined in men and women, and no studies have explored the association between dietary factors and the number of altered MetS components in the Chinese population. The aim of our study was to examine the relationship between dietary factors and the number of altered MetS components in a systematic way, using a large sample of Chinese adults (n=6034).

## METHODS
### Study population
We used the data collected by the China Health and Nutrition Survey (CHNS) 2009. The CHNS is an ongoing longitudinal survey designed to examine the association between economic and social changes and a range of health behaviours across space and time.[10] A multistage, random cluster sampling method was used to select the samples from nine provinces (Liaoning, Heilongjiang, Jiangsu, Shandong, Henan, Hubei, Hunan, Guangxi and Guizhou) in China. Questionnaire information was used to collect the data as follows: social and economic factors, health factors, nutrition factors and population factors of sociology. However, blood samples were collected only for the first time in 2009. More details about the CHNS data have been described elsewhere.[17]

In 2009, there were a total of 7755 respondents (3589 men and 4166 women) age ≥18 years. Individuals with incomplete dietary data (n=296), individuals over 75 years of age (n=419), individuals suffering from chronic diseases (hypertension, diabetes, stroke, myocardial infarction), individuals with a change in diet or physical and/or pharmacological treatment (n=906), and women in pregnancy or lactation status (n=100) were excluded. Finally, a total of 6034 adults (2800 men and 3234 women) were included in our analysis.

### Diet assessments
The 2009 CHNS combined 3 consecutive days of a 24-hour dietary recall and a household inventory to assess individual consumption. Individual dietary intake for 3 consecutive days (2 week days and 1 weekend day) was collected for every household member. Food items consumed at restaurants, canteens and other locations away from home were systematically recorded. Using food models and pictures, trained field interviewers recorded the amounts of all foods and beverage items (measured in grams) consumed during 24 hours in the previous day.[18 19] In addition, household food intake was determined on a daily basis by calculating the changes in food inventory. A Chinese balance with a maximum limit of 15 kg and a minimum limit of 20 g was used to measure household consumption by inventory changes, from the beginning to the end of each day. At the same time, all foods and condiments in the home inventory

(purchased from markets or picked from gardens, and also food waste) were carefully recorded at the start and end of each survey.[20] The full list of food groups used in the household inventory is provided online (supplementary table 1) and elsewhere.[11]

The individual daily intake value for each food item was assessed using data from the 24-hour dietary recall, which was enhanced by use of data from household measures. Additionally, edible oils and other common condiments (sugar, starch soya sauce, salt) consumed in the household by each member, were allocated based on the proportion of reference men. The amount of nutrients for each food is available from the Chinese Food Composition Table (2nd edition).[21] Per capita daily nutrients were calculated by combining both of these.

Twelve nutrients, 13 foods, the percentage of energy from protein and fat, and the percentage of protein from various foods were selected as diet indicators according to previous studies.[6 22 23]

## Other measurements

Weight, height and waist circumference were measured by trained surveyors using standard measurement techniques. Height was measured without shoes to the nearest 0.2 cm using a portable stadiometer. Weight was measured without shoes and in light clothing to the nearest 0.1 kg on a calibrated beam scale.[24] Between the lowest rib and the iliac crest in a horizontal plane, waist circumference was measured at a point midway using non-elastictape.[25] According to a standard protocol, blood pressure was measured by trained surveyors using a mercury sphygmomanometer.[26]

Blood samples were collected by vein puncture after an overnight fast. All samples were analysed in a national central laboratory in Beijing (medical laboratory accreditation certificate ISO 15189:2007) under strict quality control. Fasting plasma glucose was measured with a glucose oxidase phenol 4-aminoantipyrine peroxidase kit (Randox, Crumlin, UK) in a Hitachi 7600 automated analyzer (Hitachi Inc, Tokyo, Japan).[11 27] High-density lipoprotein cholesterol (HDL-C) and triglycerides (TG) were both measured by their corresponding reagents (Kyowa MedexCo, Ltd, Tokyo, Japan) using the glycerol-phosphate oxidase method, and the polyethylene glycol modified enzyme method, respectively. All lipid measurements were carried out on the Hitachi 7600 automated analyzer (Hitachi Inc, Tokyo, Japan).[28] History of patients (past/present) suffering from one or more chronic diseases (hypertension, diabetes, stroke, myocardial infarction), smoking and drinking (y/n), and occupational physical activity was collected using questionnaires.

## Definition of MetS and its components

The definition of MetS was based on the most recent Joint Interim Statement (JIS) criteria.[29] As mentioned earlier, subjects under treatment for chronic diseases (hypertension, diabetes, stroke, myocardial infarction) were excluded from the present study. Thus, MetS was defined as the presence of three or more of the following five components: (1) abdominal obesity, waist circumference in men ≥90 cm and women ≥80 cm, specifically for Asian adults;[6] (2) elevated blood pressure, systolic/diastolic blood pessure (SBP/DBP) ≥130/85mm Hg; (3) hypertriglyceridaemia, TG ≥1.70 mmol/L; (4) low HDL-C, <1.0 mmol/L in men and <1.3 mmol/L in women; and (5) elevated blood glucose levels, fasting blood glucose ≥5.6 mmol/L. The number of clustering MetS components was calculated by adding the number of MetS components.[6]

## Definition of covariates

According to the definition of WHO, adults who smoked at least one cigarette a day are defined as current smokers.[12] To determine alcohol consumption, individuals were asked the question: 'Have you consumed alcohol (beer, wine or other alcoholic beverage) during the past year (yes, no)?'.[2] Residence was divided into urban and rural, and geographical region was divided into the North (Liaoning, Heilongjiang, Henan and Shandong) and the South (Hunan, Hubei, Jiangsu, Guangxi and Guizhou).[12] Body mass index (BMI) was calculated as weight divided by height squared (kg/m$^2$).

Current economic status was assessed by mean per capita annual income (unit: RMB Yuan) in the year before the 2009 CHNS. The subjects were classified as low, moderate, and high income level by trisected percentiles of the mean per capita annual income. Participants were interviewed using a semi-quantitative assessment to determine their occupational, domestic, travel, and leisure physical activity levels. The intensity (metabolic equivalent, MET, unit kcal/kg$^{-1}$.h$^{-1}$) of each activity in the questionnaire was coded according to the compendium of physical activities.[30] The total metabolic equivalent (MET-hours/week) was a combined score calculated by multiplying the frequency, duration, and intensity of physical activity. Total metabolic equivalent scores were categorised into three levels (mild, moderate, and high) by trisected percentiles for further analysis.

## Statistical analysis

Statistical Analysis System 9.2 (SAS Institute, Cary, NC, USA) was used for all statistical analyses. Subjects were categorised into four groups according to the number of clustering MetS components (0, 1, 2, 3–5). Values of three to five clustering MetS components were combined because only a few subjects had alterations in four or five MetS components (four altered components: 121 men and 215 women; five altered components: 28 men and 64 women). Associations between categorical variables were tested using a $\chi^2$ test. Comparisons between continuous variables were performed by analysis of variance, and 95% confidence intervals (CI) were estimated.

Assessment of model effects and trend tests were performed to analyse the association between each of the dietary indicators and the number of altered MetS

components using a generalised linear regression model. Adjustments were made for all potential confounding factors, including age, energy intake, alcohol intake, smoking status, per capita annual income, education levels, physical activity levels, residence (urban/rural), and geographical region (North/South). Mean nutritional intakes were calculated by the number of MetS components (0, 1, 2, 3–5). The estimated marginal means for each of the dietary indicators of altered MetS components groups 1, 2, and 3–5 were compared with those of MetS components group 0 by multiple comparison test adjusted using the Bonferroni correction.

All reported p values were estimated using two-sided analysis. A value of p<0.05 was considered statistically significant.

## RESULTS
### Characteristics of the study population
table 1 shows the characteristics and distribution of all the subjects included in the study categorised by sex. Mean±SD age was 47.3±13.6 years for men and 47.6±13.1 years for women. BMI was 23.0±3.1 kg/m$^2$ for men and 23.0±3.2 kg/m$^2$ for women. The number of altered MetS components in both men and women gradually increased as age and BMI increased.

We estimated that 58.0% of men currently smoked and 63.0% consumed alcohol; 3.2% of women smoked and 9.2% consumed alcohol.

Overall, there was a positive association between the number of altered MetS components and the proportion of moderate and high per capita annual income, but only in men. In women, the number of altered MetS components was associated with a higher proportion of individuals with an educational level of primary school and below. In addition, there was a negative relationship between the number of altered MetS components in men and women and moderate and high physical activity levels.

### Prevalence of MetS and metabolic abnormalities
As shown in table 2, the overall prevalence of MetS was significantly higher in women (22.3%, 95% CI 20.9% to 23.7%) than in men (17.3%, 95% CI 15.9% to 18.7%). Significant differences between sexes were noted in the analysis of individual components. Elevated blood pressure, hypertriglyceridaemia and hyperglycaemia were more frequent in men than in women. Abdominal obesity and low HDL cholesterol were more frequent in women than in men. In men, hypertension was the most prevalent component of MetS (38.6%), followed by hypertriglyceridaemia (31.5%). In women, the most prevalent MetS component was abdominal obesity (49.4%), followed by low HDL-C (31.0%).

### Food and nutrient intake in correlation with different MetS components in men
table 3 shows multivariate adjusted mean food and nutrients intake, according to the number of altered MetS

components in men. Higher daily intake of meat (range of intake: 90.6–105.7 g) and fish/seafood (30.4–42.3 g) was positively correlated with an increased number of altered MetS components (Ptrend=0.016 for meat; Ptrend=0.001 for fish/seafood). Although analysis of covariance or trend tests did not reach statistical significance, the number of altered MetS components were positively correlated with daily intakes of abdominal organs (2.9–4.4 g; Ptrend=0.050), beans (17.0–19.5 g; Ptrend=0.113), and nuts (3.2–4.1 g; Ptrend=0.207).

Among the food groups, there was a negative correlation between the number of altered MetS components and the daily intake of cereals (453.9–440.8 g; analysis of covariance (ANCOVA) p=0.083, Ptrend=0.058), coarse cereals (16.5–12.7 g; Ptrend=0.051), and tubers (37.3–32.7 g; ANCOVA p=0.048, Ptrend=0.030).

An increased number of altered MetS components were positively correlated with a higher daily intake of protein (70.4–73.4 g; Ptrend=0.0004) and cholesterol (238.7–266.6 mg; Ptrend=0.004). It is worth noting that there was a positive correlation between an increased number of altered MetS components and an increased energy daily intake of protein (12.3–12.8%; ANCOVA p=0.001, Ptrend=0.0001), and with increased daily intake of protein of animal origin (28.6–30.4%%; ANCOVA p=0.021, Ptrend=0.020).

Among the dietary indicators, there was a marginally negative difference between the number of altered MetS components and the intake of dietary fibre (11.7–11.5 g; ANCOVA p=0.058). Dietary cholesterol intake in the groups of MetS components 1, 2 and 3–5 were significantly higher than in the group 0 (ANCOVA p=0.002; multiple comparison test p=0.006, p=0.003, p=0.012).

Further analyses were conducted between other kinds of nutrients and the number of altered MetS components in men. For more details, see online supplementary table 2.

### Food and nutrient intake in correlation with different MetS components in women
table 4 shows the multivariate adjusted mean food and nutrient intake, according to the number of MetS components in women. There was a positive correlation between the number of altered MetS and the daily intake of cereals (range of intake: 345.3–351.6 g), especially for wheat (101.9–112.6 g; Ptrend=0.066). In addition, there was a negative correlation between the number of altered MetS components and the daily intake of milk and dairy products (17.8–11.5 g; ANCOVA p=0.020, Ptrend=0.002). The daily intake of milk and dairy products for the group 3–5 of altered MetS components was significantly lower than the intake for the group 0 (multiple comparison test p=0.009).

Higher daily intake of sodium (3862.3–4005.7 mg; ANCOVA p=0.001, Ptrend=0.032) was positively correlated with an increased number of altered MetS components, respectively. However, as the daily intake

**Table 1** Subject characteristics according to the number of metabolic syndrome (MetS) components

| | Number of MetS components | | | | | | |
| | ALL | 0 | 1 | 2 | 3–5 | p Value* | *P*trend |
|---|---|---|---|---|---|---|---|
| Men n, % | 2800 (100.0%) | 866 (30.9%) | 832 (29.7%) | 618 (22.1%) | 484 (17.3%) | | |
| Age (years) | 47.3±13.6 | 44.0±14.1 | 48.0±13.7 | 48.7±12.8 | 50.2±12.0 | <0.001 | <0.001 |
| Body mass index (kg/m$^2$) | 23.0±3.1 | 21.2±2.4 | 22.5±2.8 | 24.0±2.8 | 25.7±2.9 | <0.001 | <0.001 |
| Energy intake (Kcal/day) | 2308.4±610.4 | 2349.9±614.1 | 2317.5±607.0 | 2296.7±601.3 | 2269.6±27.8 | 0.109 | 0.016 |
| Urban residents (n,%) | 740 (26.4) | 181 (20.9) | 204 (24.5) | 183 (29.6) | 172 (35.5) | <0.001 | <0.001 |
| Northern residents (n,%) | 1176 (42.0) | 300 (34.6) | 342 (41.1) | 292 (47.2) | 242 (50.0) | <0.001 | <0.001 |
| Moderate and high physical activity (n,%) | 1674 (59.7%) | 556 (19.9) | 516 (18.4) | 349 (12.5) | 253 (9.0) | <0.001 | <0.001 |
| Moderate and high per capita annual income (n,%) | 1864 (67.1%) | 564 (20.3) | 532 (19.2) | 424 (15.3) | 344 (12.4) | 0.005 | 0.001 |
| Education level (n,%) | | | | | | | |
| Primary school and below | 876 (31.3) | 253 (9.0) | 283 (10.1) | 197 (7.0) | 143 (5.1) | | |
| Junior middle school | 1122 (40.1) | 369 (13.2) | 333 (11.9) | 225 (8.0) | 195 (7.0) | 0.062 | 0.438 |
| High school and above | 798 (28.5) | 243 (8.7) | 214 (7.7) | 195 (7.0) | 146 (5.2) | | |
| Drinker (n,%) | 1764 (63.0%) | 525 (60.6%) | 523 (62.9%) | 402 (65.0%) | 314 (64.9%) | 0.267 | 0.061 |
| Current smoker (n,%) | 1623 (58.0%) | 516 (59.6%) | 496 (59.6%) | 341 (55.2%) | 270 (55.8%) | 0.189 | 0.062 |
| Women | | | | | | | |
| n, % | 3234 (100.0%) | 837 (25.9%) | 906 (28.0%) | 770 (23.8%) | 721 (22.3%) | | |
| Age (years) | 47.6±13.1 | 41.2±12.6 | 46.4±13.1 | 50.2±12.0 | 53.7±11.1 | <0.001 | <0.001 |
| Body mass index (kg/m$^2$) | 23.0±3.2 | 20.8±2.1 | 22.3±2.9 | 24.0±2.9 | 25.3±3.0 | <0.001 | <0.001 |
| Energy intake (Kcal/day) | 1963.4±535.8 | 1972.5±522.0 | 1966±532.7 | 1974±552.5 | 1938.3±537.4 | 0.549 | 0.275 |
| Urban residents (n,%) | 948 (29.3) | 233 (27.8) | 267 (29.5) | 208 (27.0) | 240 (33.3) | 0.039 | 0.068 |
| Northern residents (n,%) | 1369 (42.3) | 279 (33.3) | 388 (42.8) | 349 (45.3) | 353 (49.0) | <0.001 | <0.001 |
| Moderate and high physical activity (n,%) | 1921 (59.4%) | 553 (17.1) | 552 (17.1) | 453 (14.0) | 363 (11.2) | <0.001 | <0.001 |
| Moderate and high per capita annual income (n,%) | 2119 (66.4%) | 574 (18.0) | 581 (18.2) | 487 (15.3) | 477 (14.9) | 0.032 | 0.616 |
| Education level (n,%) | | | | | | | |
| Primary school and below | 1508 (46.7%) | 280 (8.7) | 413 (12.8) | 406 (12.6) | 409 (12.7) | | |
| Junior middle school | 1003 (31.0%) | 297 (9.2) | 306 (9.5) | 213 (6.6) | 187 (5.8) | <0.001 | <0.001 |
| High school and above | 720 (22.3%) | 259 (8.0) | 187 (5.8) | 150 (4.6) | 124 (3.8) | | |
| Drinker (n,%) | 299 (9.2%) | 90 (10.8%) | 79 (8.7%) | 64 (8.3%) | 66 (9.2%) | 0.338 | 0.243 |
| Current smoker (n,%) | 105 (3.2%) | 18 (2.2%) | 28 (3.1%) | 26 (3.4%) | 33 (4.6%) | 0.061 | 0.008 |

Values shown are mean±SD
*Statistical significance was determined by analysis of variance or $\chi^2$ test.

of β-carotene decreased (1578.6–1382.7 µg; ANCOVA p=0.001, *P*trend=0.007) the number of MetS components increased. The daily intake of β-carotene in the altered MetS components groups 1 and 3–5 was significantly lower than in group 0 (multiple comparison test p=0.001 and p=0.003).

Further analysis were conducted between other kinds of nutrients and number of altered MetS components in

**Table 2** Prevalence of metabolic abnormalities according to the number of metabolic syndrome (MetS) components

| | ALL | | Number of MetS components | | | | | | | |
| | | | 0 | | 1 | | 2 | | 3–5 | |
| | n,% | 95% CI | n,% | 95% CI | n,% | 95% CI | n,% | 95% CI | n,% | 95% CI |
|---|---|---|---|---|---|---|---|---|---|---|
| **Men** | | | | | | | | | | |
| Metabolic abnormalities (n,%) | 2800 (100.0) | 15.9 to 18.7 | 866 (30.9) | 27.8 to 34.0 | 832 (29.7) | 28.0 to 31.4 | 618 (22.1) | 20.6 to 23.6 | 484 (17.3) | 15.9 to 18.7 |
| Waist circumference (≥90 cm) | 710 (25.4) | 23.8 to 27.0 | 0(–) | — | 111 (13.3) | 11.0 to 15.6 | 254 (41.1) | 37.2 to 45.0 | 345 (71.3) | 67.3 to 75.3 |
| Triglyceride (≥1.7 mmol/L) | 883 (31.5) | 29.8 to 33.2 | 0(–) | — | 187 (22.5) | 19.7 to 25.3 | 296 (47.9) | 44.0 to 51.8 | 400 (82.6) | 79.2 to 86.0 |
| HDL cholesterol (<1.0 mmol/L) | 411 (14.7) | 13.4 to 16.0 | 0(–) | — | 56 (6.7) | 5.0 to 8.4 | 119 (19.3) | 16.2 to 22.4 | 236 (48.8) | 44.3 to 53.3 |
| Blood pressure (SBP/DBP ≥130/85 mm Hg) | 1081 (38.6) | 36.8 to 40.4 | 0(–) | — | 368 (44.2) | 40.8 to 47.6 | 354 (57.3) | 53.4 to 61.2 | 359 (74.2) | 70.3 to 78.1 |
| Fasting glucose (≥5.6 mmol/L) | 612 (21.9) | 20.4 to 23.4 | 0(–) | — | 110 (13.2) | 10.9 to 15.5 | 213 (34.5) | 30.8 to 38.2 | 289 (59.7) | 55.3 to 64.1 |
| **Women** | | | | | | | | | | |
| Metabolic abnormalities (n,%) | 3234 (100.0) | 20.9 to 23.7 | 837 (25.9) | 22.9 to 28.9 | 906 (28.0) | 26.5 to 29.5 | 770 (23.8) | 22.3 to 25.3 | 721 (22.3) | 20.9 to 23.7 |
| Waist circumference (≥80 cm) | 1599 (49.4) | 47.7 to 51.1 | 0(–) | — | 396 (43.7) | 40.5 to 46.9 | 555 (72.1) | 68.9 to 75.3 | 648 (89.9) | 87.7 to 92.1 |
| Triglyceride (≥1.7 mmol/L) | 793 (24.5) | 23.0 to 26.0 | 0(–) | — | 78 (8.6) | 6.8 to 10.4 | 209 (27.1) | 24.0 to 30.2 | 506 (70.2) | 66.9 to 73.5 |
| HDL cholesterol (<1.3 mmol/L) | 1002 (31.0) | 29.4 to 32.6 | 0(–) | — | 191 (21.1) | 18.4 to 23.8 | 297 (38.6) | 35.2 to 42.0 | 514 (71.3) | 68.0 to 74.6 |
| Blood pressure (SBP/DBP ≥130/85 mm Hg) | 964 (29.8) | 28.2 to 31.4 | 0(–) | — | 162 (17.9) | 15.4 to 20.4 | 324 (42.1) | 38.6 to 45.6 | 478 (66.3) | 62.8 to 69.8 |
| Fasting glucose (≥5.6 mmol/L) | 594 (18.4) | 17.1 to 19.7 | 0(–) | — | 79 (8.7) | 6.9 to 10.5 | 155 (20.1) | 17.3 to 22.9 | 360 (49.9) | 46.3 to 53.5 |

DBP, diastolic blood pressure; SBP, systolic blood pressure.

**Table 3** Energy and multivariate adjusted[†,‡] mean food and nutrient intake according to the number of metabolic syndrome (MetS)components in men (n=2800)

| | Number of MetS components | | | | p Value[§] | Ptrend |
|---|---|---|---|---|---|---|
| | 0 | 1 | 2 | 3–5 | | |
| n,% | 866 (30.9%) | 832 (29.7%) | 618 (22.1%) | 484 (17.3%) | | |
| Energy | 2349.9±20.8 | 2317.5±21.3 | 2296.7±24.6 | 2269.6±27.8 | 0.109 | 0.016 |
| Nutrients† | | | | | | |
| Protein (energy %) | 12.3±0.1 | 12.6±0.1 | 12.6±0.1[*] | 12.8±0.1[**] | 0.001 | 0.0001 |
| Fat (energy %) | 31.7±0.4 | 31.6±0.4 | 32.1±0.4 | 31.9±0.5 | 0.748 | 0.558 |
| Carbohydrate (energy %) | 55.8±0.4 | 55.7±0.4 | 55.1±0.4 | 55.0±0.5 | 0.460 | 0.141 |
| Nutrients‡ | | | | | | |
| Protein (g) | 70.4±0.6 | 71.5±0.6 | 72.1±0.6 | 73.4±0.7[**] | 0.006 | 0.0004 |
| Beans (protein %) | 6.6±0.3 | 6.2±0.3 | 6.4±0.3 | 6.5±0.4 | 0.849 | 0.935 |
| Animal foods (protein %) | 28.6±0.6 | 29.9±0.6 | 31.0±0.6[**] | 30.4±0.7 | 0.021 | 0.020 |
| Other vegetable food (protein %) | 62.9±0.6 | 61.1±0.6 | 60.3±0.7[*] | 60.5±0.8[*] | 0.011 | 0.007 |
| Fat (g) | 80.8±1.0 | 80.8±1.0 | 82.1±1.1 | 81.2±1.2 | 0.790 | 0.595 |
| Carbohydrate (g) | 321.0±2.3 | 319.1±2.3 | 314.7±2.6 | 317.7±2.9 | 0.263 | 0.192 |
| β-Carotene (µg) | 1493.6±42.8 | 1431.6±42.1 | 1485.1±47.4 | 1472.0±53.4 | 0.698 | 0.957 |
| Vitamin C (mg) | 78.3±1.4 | 76.2±1.4 | 76.6±1.6 | 75.6±1.8 | 0.597 | 0.275 |
| Vitamin E (mg) | 34.7±0.6 | 34.3±0.6 | 33.8±0.7 | 33.9±0.8 | 0.744 | 0.351 |
| Calcium (mg) | 369.8±5.3 | 371.8±5.3 | 371.5±5.9 | 375.9±6.7 | 0.903 | 0.484 |
| Sodium (mg) | 4545.9±79.2 | 4563.3±78.7 | 4681.6±88.5 | 4634.7±99.2 | 0.612 | 0.318 |
| Iron (mg) | 23.2±0.3 | 23.6±0.3 | 23.6±0.4 | 24.2±0.4 | 0.184 | 0.044 |
| Magnesium (mg) | 321.9±3.1 | 322.8±3 | 316.8±3.4 | 320.2±3.9 | 0.547 | 0.460 |
| Dietary fibre (g) | 11.7±0.2 | 11.3±0.2 | 11.2±0.2 | 11.5±0.2 | 0.058 | 0.419 |
| Cholesterol (mg) | 238.7±6.3 | 264.0±6.2[**] | 267.8±7.0[**] | 266.6±7.8[*] | 0.002 | 0.004 |
| Foods‡ | | | | | | |
| Cereals (g) | 453.9±4.9 | 443.1±4.9 | 437.1±5.5[*] | 440.8±6.2 | 0.083 | 0.058 |
| Rice (g) | 234.5±4.7 | 233.9±4.6 | 223.9±5.2 | 229.8±5.9 | 0.376 | 0.294 |
| Wheat (g) | 159.5±4.7 | 153.7±4.6 | 157.2±5.2 | 155.8±5.8 | 0.812 | 0.741 |
| Coarse cereals (g) | 16.5±1.4 | 16.0±1.4 | 14.1±1.6 | 12.7±1.8 | 0.267 | 0.051 |
| Tubers (g) | 37.3±1.9 | 39.1±1.9 | 32.9±2.1 | 32.7±2.4 | 0.048 | 0.030 |
| Nuts (g) | 3.2±0.5 | 3.5±0.5 | 3.4±0.5 | 4.1±0.6 | 0.599 | 0.207 |
| Beans (g) | 17.0±0.9 | 17.3±0.9 | 17.3±1.1 | 19.5±1.2 | 0.362 | 0.113 |
| Vegetables (g) | 335.4±5.6 | 327.9±5.6 | 328.7±6.3 | 320.8±7.1 | 0.408 | 0.119 |
| Fruits (g) | 54.0±3.2 | 50.0±3.2 | 46.9±3.6 | 48.5±4.1 | 0.444 | 0.212 |
| Fish/seafood (g) | 30.4±2.1 | 34.2±2.1 | 33.8±2.3 | 42.3±2.6[**] | 0.003 | 0.001 |
| Meats(g) | 90.6±2.9 | 95.7±2.8 | 105.7±3.2[**] | 98.4±3.6 | 0.003 | 0.016 |
| Abdominal organs (g) | 2.9±0.5 | 3.7±0.5 | 4.1±0.6 | 4.4±0.6 | 0.213 | 0.050 |
| Eggs (g) | 30.8±1.2 | 33.3±1.2 | 33.3±1.4 | 31.4±1.5 | 0.318 | 0.749 |
| Milk and dairy products (g) | 9.4±1.4 | 9.7±1.4 | 11.9±1.6 | 9.2±1.8 | 0.576 | 0.840 |
| Fats–oils (g) | 40.7±0.9 | 40.7±0.9 | 40.4±1.0 | 40.1±1.1 | 0.963 | 0.608 |

Values shown are mean±SE.
†Adjusted for age, alcohol intake, smoking, physical activity, per capita annual income, education level, residence (urban/rural) and geographical regions.
‡Adjusted for age, energy intake, alcohol intake, smoking, physical activity, per capita annual income, education level, residences (urban/rural) and geographical regions.
§Statistical significance was determined by analysis of covariance. Comparisons made with the number of MetS components 0 group, *p<0.05, **p<0.01.

**Table 4** Energy and multivariate adjusted mean food and nutrient intake according to the number of metabolic syndrome (MetS) components in women (n=3234)

| | Number of MetS components | | | | p Value§ | Ptrend |
|---|---|---|---|---|---|---|
| | 0 | 1 | 2 | 3–5 | | |
| n,% | 837 (25.9%) | 906 (28.0%) | 770 (23.8%) | 721 (22.3%) | | |
| Energy | 1972.5±18.8 | 1966±18.0 | 1974±19.5 | 1938.3±20.1 | 0.549 | 0.275 |
| Nutrients† | | | | | | |
| Protein (energy %) | 12.7±0.2 | 12.7±0.2 | 12.8±0.2 | 12.8±0.2 | 0.922 | 0.567 |
| Fat (energy %) | 33.2±0.6 | 33.4±0.6 | 33.4±0.7 | 33.3±0.7 | 0.979 | 0.826 |
| Carbohydrate (energy %) | 53.9±0.6 | 53.6±0.6 | 53.6±0.7 | 53.7±0.7 | 0.909 | 0.715 |
| Nutrients‡ | | | | | | |
| Protein (g) | 61.0±0.8 | 61.1±0.8 | 61.1±0.8 | 61.4±0.8 | 0.943 | 0.583 |
| Beans (protein %) | 6.9±0.6 | 7.2±0.6 | 7.5±0.6 | 7.6±0.6 | 0.449 | 0.121 |
| Animal foods (protein %) | 32.7±1.0 | 32.3±0.9 | 32.4±1.0 | 32.3±1.0 | 0.929 | 0.612 |
| Other vegetable food (protein %) | 58.6±1.0 | 59.0±1.0 | 58.3±1.1 | 58.5±1.1 | 0.840 | 0.695 |
| Fat (g) | 72.3±1.4 | 72.6±1.4 | 72.5±1.4 | 72.6±1.5 | 0.993 | 0.819 |
| Carbohydrate (g) | 259.7±3.2 | 257.7±3.2 | 258±3.3 | 258.5±3.3 | 0.866 | 0.689 |
| β-Carotene (μg) | 1578.6±68.0 | 1391.4±66.9[**] | 1476.5±69.5 | 1382.7±70.1[**] | 0.001 | 0.007 |
| Vitamin C (mg) | 75.4±2.3 | 73.9±2.2 | 76.9±2.3 | 74.0±2.3 | 0.292 | 0.857 |
| Vitamin E (mg) | 31.6±0.9 | 31.9±0.9 | 32.9±0.9 | 32.1±0.9 | 0.309 | 0.260 |
| Calcium (mg) | 353.4±8.4 | 351.7±8.3 | 355.5±8.6 | 342.8±8.7 | 0.299 | 0.225 |
| Sodium (mg) | 3862.3±116.3 | 3984.6±114.1 | 4240.8±118.7[**] | 4005.7±119.0 | 0.001 | 0.032 |
| Iron (mg) | 19.8±0.5 | 20.2±0.5 | 20.2±0.5 | 19.8±0.5 | 0.505 | 0.817 |
| Magnesium (mg) | 277.3±4.9 | 280.9±4.8 | 278.4±5.0 | 279.6±5.1 | 0.804 | 0.748 |
| Dietary fibre (g) | 10.3±0.2 | 10.6±0.2 | 10.4±0.2 | 10.3±0.2 | 0.555 | 0.814 |
| Cholesterol (mg) | 236.7±10.1 | 237.6±10.0 | 243.7±10.3 | 245.9±10.4 | 0.640 | 0.219 |
| Foods‡ | | | | | | |
| Cereals (g) | 345.3±6.7 | 347.8±6.5 | 346.4±6.8 | 351.6±6.8 | 0.440 | 0.335 |
| Rice (g) | 216.4±7.3 | 207.7±7.1 | 211.1±7.4 | 205.5±7.5 | 0.293 | 0.139 |
| Wheat (g) | 101.9±6.4 | 106.1±6.3 | 106.1±6.5 | 112.6±6.5 | 0.280 | 0.066 |
| Coarse cereals (g) | 16.1±2.1 | 15.3±2.1 | 13.5±2.2 | 15.8±2.2 | 0.440 | 0.646 |
| Tubers (g) | 39.8±2.9 | 38.0±2.9 | 38.8±3.0 | 35.2±3.0 | 0.315 | 0.112 |
| Nuts (g) | 3.7±0.6 | 3.0±0.6 | 3.8±0.7 | 3.4±0.7 | 0.304 | 0.907 |
| Beans (g) | 15.6±1.4 | 15.7±1.4 | 16.6±1.4 | 16.7±1.4 | 0.691 | 0.274 |
| Vegetables (g) | 302.2±8.3 | 303.3±8.1 | 302.6±8.4 | 298.2±8.5 | 0.884 | 0.583 |
| Fruits (g) | 77.9±6.3 | 76.1±6.2 | 70.5±6.4 | 73.5±6.5 | 0.497 | 0.268 |
| Fish/seafood (g) | 37.3±2.9 | 34.9±2.8 | 36.8±2.9 | 37.1±3.0 | 0.670 | 0.886 |
| Meats (g) | 86.6±3.7 | 85.7±3.7 | 81.0±3.8 | 84.0±3.8 | 0.275 | 0.223 |
| Abdominal organs (g) | 3.4±0.7 | 4.5±0.7 | 4.1±0.8 | 4.3±0.8 | 0.284 | 0.251 |
| Eggs (g) | 28.3±1.8 | 27.2±1.7 | 28.6±1.8 | 27.6±1.8 | 0.714 | 0.903 |
| Milk and dairy products (g) | 17.8±2.5 | 14.9±2.4 | 13.1±2.5 | 11.5±2.5[**] | 0.020 | 0.002 |
| Fats–oils (g) | 34.7±1.2 | 35.0±1.2 | 36.0±1.2 | 36.0±1.2 | 0.459 | 0.133 |

Values shown are mean±SE.
†Adjusted for age, alcohol intake, smoking, physical activity, per capita annual income, education level, residence (urban/rural) and geographical regions.
‡Adjusted for age, energy intake, alcohol intake, smoking, physical activity, per capita annual income, education level, residences (urban/rural) and geographical regions.
§Statistical significance was determined by analysis of covariance (ANCOVA). Comparisons made with the number of MetS components 0 group, *p<0.05, **p<0.01.

women. See the online supplementary table 3 for more details.

## DISCUSSION

Using data from a partially national representative sample of Chinese adults, we found that higher daily intake of meat, fish/seafood, protein, and cholesterol, and lower daily intake of coarse cereals, tubers, and dietary fibre were correlated with the number of altered MetS components in men. We also found that higher daily intake of wheat and sodium, and lower intake of β-carotene, milk and dairy products was correlated with the number of altered MetS components in women. Rather than using a dietary pattern method of analysis, it might be more accurate to evaluate the effect of the single foods or nutrients on the risk of MetS systematically.[3–6]

Our study found that the increased intake of meat and fish/seafood was positively associated with the number of altered MetS components in the Chinese male population. This correlation is closely related to the rapid evolution of the diet in China, due to the remarkable economic developments and changes in lifestyle since the 1980s. During this social revolution, the popular dietary pattern shifted away from traditional Chinese food, composed mainly of cereals, vegetables and few animal foods, to the consumption of a western diet (eg, high energy, high fat, high protein and low dietary fibre).[31] High energy density in carbohydrate-rich food is evidently linked with high levels of obesity and nutrition-related non-communicable diseases.[32]

Previous findings indicated that meat patterns of Chinese meat consumers were characterised by a predominant intake of fatty fresh pork.[33] But the high intake of fats from meat, particularly saturated fat, has been associated with higher plasma lipoprotein levels and higher blood pressure levels.[34] According to the Dietary Reference Intake (DRIs) for Chinese adults, general recommendations propose a total intake of fats of 20–30% of the daily caloric consumption with an emphasis on unsaturated fat; saturated fat should represent <10% of total ingested calories. In opposition to these general recommendations, our study showed that the intake of fats (% of energy) was slightly higher than the recommended value in both genders (eg, 31.7% in men and 33.2% in women with zero altered MetS components; 31.9% in men and 33.3% in women with three or more altered MetS components). Furthermore, our previous study revealed that, over the past two decades, adults in China consumed increasingly high amounts of cholesterol deviating from the recommended intake level.[35] However, a positive correlation between cholesterol intake and the number of MetS components was only found in men in our study. In contrast, other recent studies have indicated that the effect of dietary cholesterol on the incidence of cardiovascular disease and on the level of serum cholesterol is still unclear.[36 37] Thus, compared with a total fat intake, this observation suggests that the types of fats and cholesterol consumed may be latent factors influencing the risk of elevated MetS components in men.

Intake of dietary protein also influences the MetS components, through the percentage of protein from various foods, or through quality supplementation of specific amino acids.[38] A review of epidemiologic evidence indicated that intake of dietary animal proteins appears to increase the risk of MetS.[39] Gadgil et al found that a diet high in animal protein was associated with higher BMI, higher waist circumference, higher total cholesterol, and higher low-density lipoprotein cholesterol in South Asians living in the USA.[40] According to the DRIs for Chinese adults, recommendations for daily protein intake are 65 g for men and 55 g for women.[41] Our study showed that protein intake both in men and women was also somewhat higher than the recommended amount (eg, 70.4 g and 61.0 g per day in men and women, respectively, with zero altered MetS components, and 73.4 g and 61.4 g per day in men and women, respectively, with three or more altered MetS components). Interestingly, a positive correlation was observed in men between the intake of dietary protein, the percentage of protein ingested from animal food (protein %), and the number of MetS or alterations of its components. This observation indicated that men and women may have different sources and different composition of dietary protein, and that in men a diet rich in animal protein possibly has an important relationship with the MetS components.[38]

Park et al reported that elevated serum ferritin levels were independently associated with the future development of MetS, during a 5-year follow-up period.[42] Additionally, a previous study found that serum ferritin levels were higher in men than in women in Chinese adults, and described a positive association between ferritin levels and the prevalence of MetS in both men and women.[43] Similar to previous findings, after adjusting for potential confounding factors, we found a positive correlation between meat intake and an increased number of altered MetS components only in men. This finding indicated that a higher iron intake in men was possibly related to the number of MetS components.

High cereal fibre content and low glycaemic index are inherent attributes of most whole-grain foods.[3] Coarse cereals and tubers can provide rich dietary fibre, less fat and less protein than animal food.[12] Similar to a study of middle-aged and elderly Japanese people,[6] we found that dietary fibre intake had a marginally significant difference among altered MetS components groups 0, 1, 2 and 3–5 overall. Other data indicated that the protective role of fibre from cereals was higher than that of fibre from other sources against the development of MetS and type 2 diabetes.[3] Therefore, an inverse association between coarse cereals and tubers and the number of MetS components in our study may be partly explained by dietary fibre. Magnesium is another component of whole grains that may improve insulin sensitivity.[5] However, dietary magnesium intake in both genders was not correlated with the number of altered MetS components in the present study.

Interestingly, our study showed a marginally significant positive correlation between wheat intake and the number of MetS components only in women, after adjusting for potential confounding factors. In China, wheat and rice belong to dietary patterns with different characteristics; wheat tends to be a major component of a modern high-wheat dietary pattern, as mentioned earlier.[7–10] One explanation for this correlation is that the refining wheat process before cooking results in the removal of healthy wheat constituents (eg, fibre, vitamins, minerals, phytonutrients and essential fatty acids).[44] Thus, higher carbohydrate consumption (eg, higher intake of highly refined starch and sugar) could have adverse effects on the profile of metabolic risk.[45] In addition, high glycaemic index foods (such as noodles, bread and steamed bread) can induce inflammation, increase oxidant activities, and cause rapid fluctuations of blood glucose and insulin levels.[46] The other explanation for this correlation is that females tend to have a high intake of refined carbohydrates, most of which is prepared by wheat with high levels of salt.[9 10] A stronger positive correlation was observed between sodium intake and the number of altered MetS components in women, which may reflect female dietary habits. High sodium diet not only elevated blood pressure, but also reduced insulin sensitivity.[47]

In keeping with previous studies,[6 48–50] our data showed that the intake of β-carotene, as well as milk and dairy products, was negatively associated with the number of altered MetS components in women. Carotenoids, mainly found in fruit and fresh vegetables, can capture free radicals, quenching singlet oxygen, and improve the protective capability of antioxidants.[51] Milk, rich in calcium and vitamin D, could accelerate the elimination of abdominal fat.[49] More recently, the Chinese Nutrition Society recommended a daily intake of fruits of 200–350 g per capita per day, and a daily intake of milk or its products of 300 g per capita per day for the general Chinese population.[52] In contrast to these recommendations, the mean dietary intake of fruits and milk and dairy products was very low for men and women (eg, 54.0 g or 77.9 g of fruit per day in men and women, respectively, with zero altered MetS components, and 9.4 g and 17.8 g milk and dairy products per day in men and women, respectively, with zero altered MetS components). Therefore, our study suggests that increasing the intake of β-carotene, milk and dairy products could possibly be of great potential to prevent MetS, or the alteration of its components, in the Chinese population, especially in women.

Fruits, vegetables, whole grains, fish, nuts, and dairy products are six kinds of food that may be beneficial for preventing MetS in various dietary patterns, such as the Mediterranean dietary pattern, dietary approaches to stop hypertension, and the Nordic diet.[53] It is well known that the Mediterranean dietary pattern has important features, represented by the daily intake of whole grain, fruits, vegetables and dairy products, as well as the weekly intake of fish, poultry, nuts and beans.[54] Similar to

previous multiethnic reports, our study showed a downward trend between the intake of fruits and vegetables and the number of altered MetS components, but not with statistical significance for either men or women. However, in our study a higher intake of fish/seafood (eg, fish, shrimp, crab and shellfish) was positively correlated with an increased number of altered MetS components in men. This inconsistency may possibly be due to the existence of an 'alcohol' dietary pattern, characterised by animal foods (eg, meat, fish/seafood), alcohol and nuts in the Chinese male population,[12] by the fact that in China, fish/seafood is preserved with high levels of salt, or that the intake of fried dough with soy-bean milk is a typical breakfast for Chinese people.[11]

This study has several limitations. First, there may be residual confounding factors in our study. For instance, considerable recent progress has been made in the identification of genetic loci that are associated with insulin resistance and MetS.[55] MetS has been related to the increasing prevalence of obesity, which is escalating even among older people.[56] Hormone therapy improves the lipid and oxidative alterations that occur in MetS in post-menopausal women.[57] Cumulative alcohol consumption may be an especially important confounding factor. Our previous study found that an 'alcohol' dietary pattern was present in Chinese men.[12] In this study, the overall prevalence of alcohol consumption was 64.9% in men and 9.2% in women with three or more altered MetS components. Moderate alcohol intake has been consistently associated with a lower risk of fatal and non-fatal cardiovascular disease. Wine is an important component of the Mediterranean diet,[44] but heavy drinking was positively associated with MetS and with alterations of its components.[58] Due to lack of relevant information, the important factors mentioned above were excluded from our analysis, which may limit the strength of our findings.

Second, the mixed intake of food and nutrients may create more complex interactive effects than the analysis of the intake of foods and nutrients as indicators.[14] For instance, previous findings have shown that the 'Green Water' dietary pattern, which is characterised by a high intake of rice and vegetables, and a moderate intake of animal foods, has beneficial effects on MetS or on its components.[7–9] There was no significant negative correlation between the intake of rice and vegetables and the number of altered MetS components in both genders in the present study.

Third, in this study, we measured diet using a combination of 3 consecutive days by 24-hour dietary recall, and a household inventory. This method has a few shortcomings. One is that the allocation of oil and other condiments among household members was based on the proportion of reference men, instead of total food intake, which could lead to an additional level of error for the analysis of individual intake. Another shortcoming is that dietary data collected by this method omits adjustment for intake of oil and other condiments for foods eaten

away from home.[20] Additionally, it is unclear whether short-term records of dietary recall can fully reflect the long-term intake of a dietary pattern.[59]

Fourth, despite a progressive increase or decrease in intake levels of some foods and nutritional factors (eg, meat, protein, cholesterol in men and β-carotene, milk and dairy products in women) according to the cumulative distribution of MetS components, it is uncertain whether these levels would have a direct impact on MetS and their metabolic profiles. Possible reasons for this uncertainty is due to our cross-sectional design, and we could not come to any causal conclusion to explain the relationships between dietary factors and the number of altered MetS components.

Fifth, although some correlations were observed between dietary factors and the number of altered MetS components among Chinese adults aged 18–75 years in the present study, it is difficult to predict an overall and long-term association between dietary factors and the number of altered MetS components for the Chinese adult population. On the one hand, our study is just a partial national representative sample including only nine provinces of China, and prior correlation findings have not previously been reported to the best of our knowledge. On the other hand, a gradual increase in the level of urbanisation has led to improvement in the dietary intake of the Chinese adult population over the past decade. It is necessary to conduct multicentre studies that represent the groups of interest in different regions and periods, and thereby extend the applicability of such dietary factors in the battle against MetS in China.

## CONCLUSION

Our study showed that some foods and nutritional factors are correlated with the number of altered MetS components among the Chinese adult population aged 18–75 years. However, whether consumption of these foods and nutritional factors would directly affect metabolic disorders requires further investigation; further prospective and clinical research is needed to examine these associations.

**Contributors** MWC, BZ conceived the study. MWC, YFOY and WWD completed all statistical analyses and interpretation of data. MWC drafted the manuscript. ZHW and WWD contributed to the discussion. HJW and BZ revised the manuscript. All authors contributed to the paper and the final manuscript. All authors read and approved the final manuscript.

**Funding** The work was supported by the National Institutes of Health(NIH: R01-HD30880,DK056350 and R01-HD38700).

**Competing interests** None declared.

**Patient consent** Obtained.

**Ethics approval** All protocols, instruments and the process for obtaining informed consent for this study were approved by the Ethical Committee of the University of North Carolina at Chapel Hill and the National Institute for Nutrition and Food Safety of USA, Chinese Center for Disease Control and Prevention. Signed consent forms were obtained from all participants.

**Provenance and peer review** Not commissioned; externally peer reviewed.

**Data sharing statement** The data used for the analysis are available from the CPC at http://www.cpc.unc.edu/projects/china.

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
