## [Reviewer comments · BMJ Open]

ARTICLE DETAILS

TITLE (PROVISIONAL)	Relationship between dietary factors and the number of altered metabolic syndrome components in Chinese adults: a cross-sectional study using data from the China Health and Nutrition Survey
AUTHORS	Cheng, Maowei; Wang, Huijun; Wang, Zhihong; Du, Wenwen; Ouyang, Yifei; Zhang, Bing

VERSION 1 - REVIEW

REVIEWER	Edyta Suliga Faculty of Medicine and Health Sciences, Jan Kochanowski University, Kielce, Poland
REVIEW RETURNED	28-Nov-2016

GENERAL COMMENTS	In this work, the authors analyzed the relationship between dietary factors and the number of altered metabolic syndrome components in adults. The analyses were based on cross-sectional data from 6034 adults aged 18 or older. Generally, the research is meaningful for public health institutions since the findings may help to identify the associations between dietary patterns and the occurrence of metabolic syndrome, which has not been fully explained yet. In my opinion, the manuscript is worth publishing providing the authors make amendments according to the following suggestions: • In the method description (page 5: Diet assessments), it is written that the data regarding the dietary patterns were collected on the basis of a 24-hour dietary recall, and a household inventory. However, the authors do not explain precisely if and what data from household inventory were used in the analysis.• How did the authors define the size of a portion of products and dishes in the case of 24-hour dietary recall?• The authors say that nutrients and kinds of food were chosen as diet indicators according to previous studies (items 6,21 and 22 in the bibliography). Those studies were done on different populations though, in which completely different kinds of food and nutrients could have an influence on the occurrence of MetS. It is worth checking whether in the populations in question, other kinds of food and nutrients that may have a correlation with MetS can be identified.• We can learn from the chapter entitled 'Definition of covariates' (page 7) that many significant confounders were not taken into account. Many studies show that a history of smoking, the status of menopause in women, as well as undergoing hormone therapy may influence the occurrence of MetS and its components (look at Suliga E. et al. PLoS ONE 2016; 11(4): e0154511.). It should be mentioned in 'study limitations'.• The authors say (Statistical analysis, page 8) that: 'mean nutritional
--

	intakes were calculated by the number of MetS components (0, 1, 2, 3-5). The estimated means for each of the dietary indicators of altered MetS components groups 1, 2, and 3-5 were compared with those of MetS components group 0'. We mustn't forget that many factors influence the occurrence of MetS, including genetic ones, not only diet. It is important to mention in the discussion, to which degree the dietary patterns of the participants in the '0 components of MetS' group were in accordance with the dietary recommendations.  • Item 6 in the bibliography – wrong names of the authors. • It is recommended to update the bibliography. Only one item is from the year 2016 and it is 'Chinese Dietary Guide' and one from 2015. Others are older.
--	--

REVIEWER	Andrew Frugé University of Alabama at Birmingham, USA
REVIEW RETURNED	02-Jan-2017

GENERAL COMMENTS	This is a cross-sectional study of the relationship between diet and metabolic syndrome using the China Health and Nutrition Survey. This data is important, however additional analyses would provide greater benefit to the existing Chinese dietary and disease literature. Additionally, some of the conclusions are overstated, and care should be taken to determine whether some statistically significant findings are clinically significant. There are numerous grammatical and punctuation errors that need to be corrected. Page 2. Setting. Include "China Health and Nutrition Survey" and year of data collection (2009) Outcomes. How is the household food inventory included in analysis? This is also unclear in the methods. Results. Add spaces before opening parenthesis and after commas. Change trend test P to P trend and indicate what the comparison is for each food group or nutrient, i.e. zero MetS components compared to 3-5 components. Also, the authors chose not to report the significantly lower caloric intake of men with more MetS components (p trend = 0.016) but reported non-significant results for women, i.e. wheat. Conclusions. Add "self-reported" between the first two words. The word "prevent" is not warranted for this study. What is a healthy source of fats? You only reported total fat. Your results suggest men with more MetS components eat more meat and seafood, and eat less calories, grains, tubers and fiber. For women, higher sodium and lower beta carotene and dairy are associated with more MetS components. Page 3. Strengths and limitations. Again, how is the household inventory used in this analysis? Introduction. Line 29 change were to was. Line 40 Metabolic syndrome requires 3 of 5 components. Page 4. Dietary patterns are discussed, however the second paragraph (line 35) suggests the literature on food patterns and
---

individual nutrients do not match. It seems as though this data set provides an opportunity to unify the two approaches. Line 40 why is this sentence included and how do the two cited studies support this? "It should be noted that the idea that the Chinese traditional dietary structure is dominated by cereals should not be over emphasized"

Page 5. Methods. Is this a longitudinal survey or an ongoing cross-sectional survey? Line 37. Change the first sentence to read: "In 2009, there were a total of 7,755 respondents..." Add spaces before opening parenthesis.

Diet assessments. How were the three 24-hour recalls analyzed? Was one of the days a weekend day? Was the household inventory used in the analysis? How? Line 56 change ratio to percentage.

Page 6. Line 1 eliminate "kinds of" before nutrients and foods and change ratio to percentage.

Page 7. Definition of covariates. Change cigarettes to cigarette, change was to were (both instances), change alcoholics to alcoholic beverages, change residences to residence.

Statistical analysis. Group allocation is appropriate, however the 3-5 components group is technically those that actually have MetS. Why did you not compare intake of 0-2 components combined to 3-5 components combined (individuals without vs. with MetS)? Line 56. Was analysis of variance or analysis of covariance used?

Page 8. Results. Line 30 omit to. Line 32 omit the last and. Line 43 Please reinterpret the education chi-square results. It appears that higher education is associated with less MetS components. For men, the data so not seem to support the p values reported regarding income.

Page 9. Line 15. Define the numbers you are reporting in parenthesis as range of intake. For meat, the highest value (105.7) was consumed by men that did not have MetS (2 components), yet those with MetS consumed significantly less than 105.7g. Why are food groups mentioned that were not significant? Is visceral meat that same as abdominal organs? Please be consistent. Is a 1.5 gram difference (6-14 calories) in intake of visceral meat as part of a 2300 calorie diet relevant? Line 27, change dietary indicators to food groups. Line 38. Is it worth noting that 0.5% of energy from protein (3 grams, 12 calories). Line 46. There was a 0.5 gram range of fiber intake. Why is this reported differently from other data and why is it mentioned? Line 49 Add dietary before cholesterol and change levels to intake. Why are multiple comparison p values reported instead of ANCOVA and p trend values?

Page 10. Lines 10 and 19. Why the use of multiple comparison p value instead of ANCOVA and p trend values? Are these difference clinically significant?

Discussion. Line 40 suggests a dose-effect relationship analysis was

	used, however none was conducted. Page 11. .Add spaces before opening parenthesis. Line 4. “Higher daily intake of cholesterol from visceral meat increases the risk of cardiovascular disease” is not supported by your data. Several types of seafood have substantially higher cholesterol content and dietary cholesterol has found to have no relationship with cardiovascular disease. Line 10. Omit the before MetS. Line 42. The data in parenthesis are not a range. Line 53. Red meat is not a category in your analysis (unless poultry was omitted) and this relationship is not supported by your results. Page 12. Line 6. Your data do not support the last sentence of this paragraph. Men with MetS did not consume more iron than men in the other categories, though the trend P was significant. Line 10. This discussion is not supported by your data, especially since drinking was a binary variable. The last sentence (line 27) suggests a “strong” relationship between meat intake and MetS components, however the relationship between MetS and seafood is much “stronger.” Line 51. You cannot conclude that fiber reduced the risk of MetS in your sample. You cannot infer causality. Page 13. The discussion on wheat refers to dietary pattern analyses. Since you report wheat and fiber, but not whole wheat vs. refined, this discussion is irrelevant. Perhaps a better explanation of the coarse cereals food group in the methods might contribute to the relevance here. Additionally, the discussion about frying oils is not supported by the results. Page 14. The previous paragraph discussed the dairy and beta carotene results, yet the paragraph starting on line 6 is contradictory and should be omitted. Line 23. Change were to are. Line 33. This statement is not supported. Statistical significance is a requirement for stating a relationship. Page 15. Conclusion. You cannot make any statements regarding moderate alcohol intake as this was not measured. Again, you infer causality by suggesting your data support diet-disease relationships that would be critical to reverse for prevention of MetS. Healthy sources of fat were not measured or determined in this analysis. Refined carbohydrates were not measured in this study. Why is this in the conclusion?
--	--

VERSION 1 – AUTHOR RESPONSE

Replies to Reviewer 1

Specific Comments

1. In the method description (page 5: Diet assessments), it is written that the data regarding the dietary patterns were collected on the basis of a 24-hour dietary recall, and a household inventory. However, the authors do not explain precisely if and what data from household inventory were used in

the analysis.

Answer: Several sentences have been added in 'Diet assessments'(page 5, paragraph 4 and page 6, paragraph 1-3) and Discussion(page 20, paragraph 2) in the revised version to address this issue. Supplemental Table 1 has provided the full list of food groups using by a household inventory.

2. How did the authors define the size of a portion of products and dishes in the case of 24-hour dietary recall?

Answer: Several sentences have been added in the Diet assessments (page 5, paragraph 4, Line 52-54) to address this issue.

3. The authors say that nutrients and kinds of food were chosen as diet indicators according to previous studies (items 6,21 and 22 in the bibliography). Those studies were done on different populations though, in which completely different kinds of food and nutrients could have an influence on the occurrence of MetS. It is worth checking whether in the populations in question, other kinds of food and nutrients that may have a correlation with MetS can be identified.

Answer: Yes, we analyzed the relationship between almost all meaningful categories of food groups and MetS components in manuscript, however most food groups had a high proportion of non-consumers possibly due to the fact that dietary intake was measured over a 3 d period, and food groups consumed by fewer than 5% of our participants were not included in the dietary intake analysis. Further analysis have been conducted between other kinds of nutrients and the number of MetS components again. But we can not found any significant associations that between other kinds of nutrients and the number of MetS components in additionally analyses. For more details, see the supplementary table 2 and table 3.

4. We can learn from the chapter entitled 'Definition of covariates' (page 7) that many significant confounders were not taken into account. Many studies show that a history of a smoking, the status of menopause in women, as well as undergoing hormone therapy may influence the occurrence of MetS and its components (look at Suliga E. et al. PLoS ONE 2016; 11(4): e0154511.). It should be mentioned in 'study limitations'.

Answer: Several sentences have been added in 'study limitations' (page 20, paragraph 3) to address this issue. However, it is a pity that we are unable to obtain above-mentioned documents (Suliga E. et al. PLoS ONE 2016; 11(4): e0154511.), and we address this issue using other relative articles(Bibliography item 55-58).

5. The authors say (Statistical analysis, page 8) that; 'mean nutritional intakes were calculated by the number of MetS components (0, 1, 2, 3-5). The estimated means for each of the dietary indicators of altered MetS components groups 1, 2, and 3-5 were compared with those of MetS components group 0'. We mustn't forget that many factors influence the occurrence of MetS, including genetic ones, not only diet. It is important to mention in the discussion, to which degree the dietary patterns of the participants in the '0 components of MetS' group were in accordance with the dietary recommendations.

Answer: Several sentences have been added in 'discussion' (page 16, paragraph 3,line 45; page 17, paragraph 2,line 19; page 18, paragraph 3,line 55;) to address this issue.

6. Item 6 in the bibliography – wrong names of the authors.

Answer: Correction has been made in the revised version.

7. It is recommended to update the bibliography. Only one item is from the year 2016 and it is 'Chinese Dietary Guide' and one from 2015. Others are older.

Answer: Several latest bibliographies have been cited in 'references' in revised version(Bibliography18, 19, 20, 33, 35, 36, 37, 49, 54, 55, 56, 57, 58, 59), meanwhile the corresponding older bibliographies have been replaced or removed in revised vison.

Replies to Reviewer 2

Specific Comments

1. Overall—

Additional analyses would provide greater benefit to the existing Chinese dietary and disease literature.

Answer: Yes, additional analyses were conducted with reference to the literature on Chinese dietary and disease (page 16, paragraph 3).

Additionally, some of the conclusions are overstated, and care should be taken to determine whether some statistically significant findings are clinically significant.

Answer: Yes, a cautious and precise conclusion was given by rewording the results (Abstract, conclusion; page 17, paragraph 1 line 3; page 17, paragraph 2 line 29-31 and paragraph 3 line 46-47; Page 18, paragraph 1 line 2-5 and paragraph 2 line 31-38; page 19, paragraph 1 line 3-7 and paragraph 2 line 19-21 and line 27; page 20, Conclusions), and several sentences have been added in 'study limitations' (page 20, paragraph 4) to address the clinical significance.

There are numerous grammatical and punctuation errors that need to be corrected.

Answer: Correction has been made in the revised version.

2. Page 2.—

Setting. Include "China Health and Nutrition Survey" and year of data collection (2009)

Answer: Correction has been made in the revised version.

Outcomes. How is the household food inventory included in analysis? This is also unclear in the methods.

Answer: Several sentences have been added in 'Diet assessments' (page 5, paragraph 4 and page 6, paragraph 1-3) and Discussion (page 20, paragraph 2) in the revised version to address this issue.

Supplemental Table 1 has provided the full list of food groups using by a household inventory.

Results. Add spaces before opening parenthesis and after commas. Change trend test P to P trend and indicate what the comparison is for each food group or nutrient, i.e. zero MetS components compared to 3-5 components.

Answer: Correction has been made in the revised version.

Also, the authors chose not to report the significantly lower caloric intake of men with more MetS components (p trend=0.016) but reported non-significant results for women, i.e. wheat.

Answer: Since energy is related to numerous foods and nutrients in diet, it could be more difficult to analyze the relationship between the intake of energy and the number MetS components. Thus, we only take the intake of energy as a compound factor to adjust in analysis. However, a few prior findings provided an important clue which indicated a possible association between wheat intake and MetS, and it is necessary to further study though only a marginal significant upward trend between wheat intake and the number of MetS components in our study.

Conclusions. Add "self-reported" between the first two words. The word "prevent" is not warranted for this study. What is a healthy source of fats? You only reported total fat. Your results suggest men with more MetS components eat more meat and seafood, and eat less calories, grains, tubers and fiber. For women, higher sodium and lower beta carotene and dairy are associated with more MetS components.

Answer: Dietary intake was measured using a combination of 3 day period with 24-hour and household food inventory, so the intake of food and nutrients included the self-reported part and the weighted part. Meanwhile, we revised this part to give a more cautious conclusion.

3. Page 3.—

Strengths and limitations. Again, how is the household inventory used in this analysis?

Answer: Several sentences have been added in 'Diet assessments' (page 5, paragraph 4 and page 6, paragraph 1-3) and Discussion (page 20, paragraph 2) in the revised version to address this issue.

Supplemental Table 1 has provided the full list of food groups using by a household inventory. Introduction. Line 29 change were to was.

Answer: Correction has been made in the revised version.

Line 40 Metabolic syndrome requires 3 of 5 components.

Answer: Correction has been made in the revised version.

4. Page 4.—

Dietary patterns are discussed, however the second paragraph (line 35) suggests the literature on food patterns and individual nutrients do not match. It seems as though this data set provides an opportunity to unify the two approaches. Line 40 why is this sentence included and how do the two cited studies support this? “It should be noted that the idea that the Chinese traditional dietary structure is dominated by cereals should not be over emphasized”

Answer: Actually, due to different study design, study methods, study sample and confounding factors, till now the association between MetS components with both food patterns and individual nutrients do not agree to each other from a certain extent in Chinese population. For instance, we can draw from inconsistent conclusions from the two cited studies. However, in this study we aim was to examine the relationship between dietary factors and the number of altered MetS components in a systematic way, and it was difficult though this data to unify the two approaches. Thus, with consideration above-mentioned suggestion, this content has been removed in the revised version.

5. Page 5.—

Methods. Is this a longitudinal survey or an ongoing cross-sectional survey?

Answer: A longitudinal survey. The CHNS was initiated in 1989, with a focus on assessing the relationships between the social and economic transformation in China and the resulting effects on the health and nutritional status of the Chinese population. The corresponding original content has been streamlined in the revised vision(page 5, paragraph 2, line 13-16).

Line 37. Change the first sentence to read: “In 2009, there were a total of 7,755 respondents...” Add spaces before opening parenthesis.

Answer: Correction has been made in the revised version.

Diet assessments. How were the three 24-hour recalls analyzed? Was one of the days a weekend day? Was the household inventory used in the analysis? How?

Answer: Several sentences have been added in ‘Diet assessments’(page 5, paragraph 4 and page 6, paragraph 1-3) in the revised version to address this issue. Supplemental Table 1 has provided the full list of food groups using by a household inventory.

Line 56 change ratio to percentage.

Answer: Correction has been made in the revised version.

6. Page 6.—Line 1 eliminate “kinds of” before nutrients and foods and change ratio to percentage.

Answer: Correction has been made in the revised version.

7. Page 7.—

Definition of covariates. Change cigarettes to cigarette, change was to were (both instances), change alcoholics to alcoholic beverages, change residences to residence.

Answer: Correction has been made in the revised version.

Statistical analysis. Group allocation is appropriate, however the 3-5 components group is technically those that actually have MetS. Why did you not compare intake of 0-2 components combined to 3-5 components combined (individuals without vs. with MetS)?

Answer: Refer to large number of domestic and abroad related literature, we found numerous of studies on dietary or nutrients on individuals without vs. with MetS and it is necessary to analyze from a different point of view.

Line 56. Was analysis of variance or analysis of covariance used?

Answer: Using analysis of variance.

8. Page 8.—

Results. Line 30 omit to.

Answer: Correction has been made in the revised version.

Line 32 omit the last and.

Answer: Correction has been made in the revised version.

Line 43 Please reinterpret the education chi-square results. It appears that higher education is associated with less MetS components. For men, the data so not seem to support the p values reported regarding income.

Answer: Aim at the above-mentioned problems, it is pity that we have possibly made two unconscious errors in primary analysis by checking the original data: One is that the data of Moderate and high per capita annual income and education level was reversed in men, the other is that constituent ratios, which include three variables(Moderate and high physical activity, Moderate and high per capita annual income and Education level), has taken an inappropriate form. We partially revised the results of the analysis immediately and the amendments are highlighted in red in revised tables marked copy, meanwhile, a little changing of expression way for corresponding content has been made in the revised version(page 9, paragraph 1 line 6-7). As an important lesson, we examine the rest of the original analysis results and found no similar problems.

9. Page 9.—

Line 15. Define the numbers you are reporting in parenthesis as range of intake.

Answer: Correction has been made in the revised version.

For meat, the highest value(105.7) was consumed by men that did not have MetS (2 components), yet those with MetS consumed significantly less than 105.7g. Why are food groups mentioned that were not significant?

Answer: There was an significant difference and upward trend for overall consumption of meat in men with different MetS components(ANCOVA $P = 0.003$, trend $P = 0.016$). However we aim to compare other groups with MetS components group 0 using multiple comparison method, therefore, a comparison between MetS components group 2 and MetS components group was not conducted. Is visceral meat that same as abdominal organs? Please be consistent. Is a 1.5 gram difference (6-14 calories) in intake of visceral meat as part of a 2300 calorie diet relevant?

Answer: Yes, correction has been made in the revised version. We are uncertain about the practical significance on 1.5 gram difference for intake of visceral meat, and several sentences have been added in 'study limitations' (page 20, paragraph 4) to address this issue.

Line 27, change dietary indicators to food groups.

Correction has been made in the revised version.

Line 38. Is it worth noting that 0.5% of energy from protein(3 grams, 12 calories).

Answer: We are uncertain about this point, and several sentences have been added in 'study limitations' (page 20, paragraph 4) to address this issue.

Line 46. There was a 0.5 gram range of fiber intake. Why is this reported differently from other data and why is it mentioned ?

Answer: The inverse association between whole-grain intake and metabolic syndrome was largely explained by cereal fiber, and a significant association was no longer observed between whole-grain intake and the risk of metabolic syndrome after adjusting for cereal fiber(an excerpt from bibliography 3). Additionally, in our study the intakes of coarse cereals and tubers, which can provide rich dietary fiber, were both negatively associated with the number of altered MetS components. However, among possible related nutrients(fiber, vitamins, minerals) with coarse cereals and tubers, only a marginal significant difference between dietary fiber intake and the number of altered MetS components was showed in men. The daily intake of dietary fiber in the altered MetS components groups 1 and 2 MetS was significantly lower than in group 0 marginally also(multiple comparison test $P = 0.087$ and $P = 0.055$). Thus, we taken into account the dietary fiber content as important latent influencing factor affecting MetS risk in men.

Line 49 Add dietary before cholesterol and change levels to intake. Why are multiple comparison p values reported instead of ANCOVA and p trend values?

Answer: Yes, Correction has been made in the revised version. Since a stronger association between a higher intake of cholesterol and the number of MetS components using ANCOVA and p trend test (ANCOVA $P = 0.002$, trend $P = 0.004$). However, ANCOVA and p trend test could only conduct a statistical analysis of the overall situation, and it is difficult to detect whether there is a difference between different groups. Thus, we used multiple comparison to compare marginal means for each of dietary indicators of altered MetS components groups 1, 2, and 3-5 to those of MetS components group 0 adjusted by Bonferroni method, which was also conducted using generalized linear regression process.

10. Page 10.—

Lines 10 and 19. Why the use of multiple comparison p value instead of ANCOVA and p trend values? Are these difference clinically significant? Discussion.

Answer: The explanation is identical to the Page 9 Line 49 above-mentioned, and several sentences have been added in 'study limitations' (page 20, paragraph 4) to address the clinical significance. Line 40 suggests a dose-effect relationship analysis was used, however none was conducted.

Answer: Dose-effect relationship usually refers to a kind of exotic compounds dose and individual or group appear some effect of quantitative intensity, this concept is used here may not be precise enough. Therefore, correction has been made in the revised version.

11. Page 11.—

Add spaces before opening parenthesis.

Answer: Correction has been made in the revised version.

Line 4. "Higher daily intake of cholesterol from visceral meat increases the risk of cardiovascular disease" is not supported by your data. Several types of seafood have substantially higher cholesterol content and dietary cholesterol has found to have no relationship with cardiovascular disease.

Answer: Correction has been made in the revised version (page 16, paragraph 3).

Line 10. Omit the before MetS.

Answer: Correction has been made in the revised version.

Line 42. The data in parenthesis are not a range. Line 53. Red meat is not a category in your analysis (unless poultry was omitted) and this relationship is not supported by your results.

Answer: Correction has been made in the revised version. (page 16, paragraph 3).

12. Page 12.—

Line 6. Your data do not support the last sentence of this paragraph. Men with MetS did not consume more iron than men in the other categories, though the trend P was significant.

Answer: Correction has been made in the revised version. (page 17, paragraph 3, line 42-47).

Line 10. This discussion is not supported by your data, especially since drinking was a binary variable.

Answer: This issue have been changed to address in 'study limitations' (page 19, paragraph 3).

The last sentence (line 27) suggests a "strong" relationship between meat intake and MetS components, however the relationship between MetS and seafood is much "stronger."

Answer: In fact, this contradiction between MetS and seafood has been interpreted in other parts of the article. (page 19, paragraph 1, line 23-34).

Line 51. You cannot conclude that fiber reduced the risk of MetS in your sample. You cannot infer causality.

Answer: Correction has been made in the revised version. (page 18, paragraph 1, line 2-10).

13. Page 13.—The discussion on wheat refers to dietary pattern analyses. Since you report wheat and fiber, but not whole wheat vs. refined, this discussion is irrelevant. Perhaps a better explanation

of the coarse cereals food group in the methods might contribute to the relevance here. Additionally, the discussion about frying oils is not supported by the results.

Answer: Correction has been made in the revised version. (page 18, paragraph 2).

14. Page 14.—

The previous paragraph discussed the dairy and beta carotene results, yet the paragraph starting on line 6 is contradictory and should be omitted.

Answer: This part has been deleted in the revised version.

Line 23. Change were to are.

Answer: Correction has been made in the revised version.

Line 33. This statement is not supported. Statistical significance is a requirement for stating a relationship.

Answer: Correction has been made in the revised version.

15. Page 15.—Conclusion. You cannot make any statements regarding moderate alcohol intake as this was not measured. Again, you infer causality by suggesting your data support diet-disease relationships that would be critical to reverse for prevention of MetS. Healthy sources of fat were not measured or determined in this analysis. Refined carbohydrates were not measured in this study. Why is this in the conclusion?

Answer: Correction has been made in the revised version. (page 20, Conclusion).

VERSION 2 – REVIEW

REVIEWER	Edyta Suliga Faculty of Medicine and Health Sciences, Jan Kochanowski University, Poland
REVIEW RETURNED	31-Jan-2017

GENERAL COMMENTS	I accept this article in its present form
---